# Molecular and microscopic detection of natural and experimental infections of *Toxocara vitulorum* in bovine milk

**Amira Dewair, Mohamed Bessat**ⓘ*

Department of Parasitology, Faculty of Veterinary Medicine, Alexandria University, Alexandria, Egypt

* mohamed.bessat@alexu.edu.eg

**Data Availability Statement:** All relevant data are within the manuscript and its Supporting Information files.

**Funding:** The study was jointly supported financially by the Vmerge FP7 grant (613996) from

## Abstract

*Toxocara vitulorum* is an Ascarid nematode infecting the small intestine of buffalo and cattle particularly neonate calves, with the postnatal route through milk is the main infection source. However, little is known about shedding rates and the optimum detection methods of *T. vitulorum* larvae in the milk of the infected bovine hosts. In this study, we aimed to evaluate the use of two methods, microscopy and PCR, and their detection limits both under the experimental and natural infection situations. In doing this, *T. vitulorum* eggs extracted from naturally occurring adult female worms were successfully subjected to experimental embryonation, and larvae were implemented in experimental infection of milk in ascending infection doses of 0, 1, 5, 10, 20, 50 larvae/2-ml milk samples. With the except of negative control, microscopy-based examination detected larvae in all samples, albeit with means, ranges, and the total number of larvae were detected in exponential rates relative to larvae densities in milk samples. PCR technique corresponded well to microscopy in detecting genomic DNA of *T. vitulorum* larvae in all milk samples down to a single larva/sample. On the other hand, and by applying the same methodology approach on 50 naturally-occurring bovine colostrum/milk samples, 13 (26%) and 20 (40%) samples were tested positive for *T. vitulorum* infection by microscopy and the PCR-based detection, respectively. Of these, 11 out of 26 buffalo samples (42.30%) and 2 out of 24 cow samples (8.33%) were tested positive by microscopy, while 16 (61.54%) and 3 (12.50%) of buffalo and cow samples were tested positive by PCR, respectively. By applying the Agreement Coefficient, substantial agreement (0.77) between molecular and microscopy detection was detected from all tested samples. In conclusion, larvae of *T. vitulorum* were unequivocally detected by microscopy and molecular methods in milk samples both under the experimental and natural field situations. Nevertheless, slightly higher rates by PCR than microscopy were obtained when detecting naturally-infected milk samples. To the best of our knowledge, this is the first in situ detection of larvae of *T. vitulorum* in the milk of the naturally infected animals.

the EU, and the highly-impacted publication support fund from Alexandria University. Funders had no role in study design, data collection and analysis, decision to publish, or preparation of the manuscript. There was no additional external funding received for this study.

**Competing interests:** The authors have declared that no competing interests exist.

## Introduction

*Toxocara vitulorum* (*T. vitulorum*) is a species of Ascaridae nematodes that inhabits the small intestine of bovine animals including buffaloes, cattle, and zebu, and is especially common in the tropical and subtropical geographical areas [1]. Infections were also commonly encountered in the temperate climatic areas such as in Italy [2], France [3], Greece [4], Belgium [5], the Netherlands [6], and the UK [7]. T. vitulorum was recorded in cattle from different geographical areas spanning Africa, East Asia, the Indian subcontinent, Europe, and the North America, with various prevalence rates as 34.1% in Nepal [8], 20% in Egypt [9], 40% in the United States of America [10], 12.4% in Cambodia [11], 12% in Canada [12], 16.7% in India [13], 22.2% in Turkey [14], 0.77% in France [15], and 7.6% in Mali [16].

Usually, infection with *T. vitulorum* is subclinical, even though heavy infections with a large number of worms result in severe enteritis and diarrhea, causing considerable morbidity and mortality particularly in the age group of 1–3 months old cattle and buffalo calves [1,17]. Without proper diagnosis (usually misdiagnose with other diarrhea-causing viral, bacterial, and protozoan pathogens) and adequate treatment, high fatality rates in bovine claves cause serious economic losses.

When compared to related *Toxocara* species *T. canis* and *T. cati*, the principal agents of somatic (visceral and ocular) larva migrans in humans [18–20], *T. vitulorum* constitutes the least zoonotic significance of the three species, despite records of somatic larval migration in the experimental animal models [21,22]. In the same comparison theme, the transmission biology of *T. canis* and *T. cati* remain the most studied to date, with little focus was given to that of *T. vitulorum*.

Contaminated unpasteurized milk along with undercooked meat is the main source of human infection by *Toxocara* species larvae, specifically *T. canis* and *T. cati* [23]. Ingestion of eggs of *T. canis* in the contaminated food, and ingestion of larvae in the undercooked or raw meat of paratenic hosts and in the unpasteurized milk represents the main source of human infection [23–27]. A case of congenital ocular toxocariasis due to *T. canis* larvae has been reported in a premature child, which supported the congenital transmission route [28]. On the other hand, ingestion of eggs has barely any role in the transmission of *T. vitulorum* as larvae undergo trans-somatic migration in infected dams reaching to the mammary glands, and thus to the milk [17]. Therefore, the galagtogenic (via milk) transmission of *T. vitulorum* has been reported as the main source of toxocariasis cycling between newborn calves and dams of cattle and water buffaloes [1,29]. Moreover, the presence of *T. vitulorum* larvae in milk represents a risk factor for visceral larvae migrans, due to ingestion of unpasteurized milk of the infected animals [30]. Magnaval [31] added that toxocariasis is widespread between children who usually drink colostrum in a bad habitual manner with no pasteurization or heat treatment.

Thus diagnostic method/methods are essentially required for the proper detection of larvae of *Toxocara* species in milk, since minimizing the risk of the galagtogenic transmission. Up to date, only two reports dealing with detection of larvae of *T. canis* in the milk were published, with one reported on the detection limit of larvae in experimentally contaminated bovine milk [32], while the other on detecting larvae in the milk of experimentally infected rabbit as a paratenic host [20], with both studies had applied the centrifuge-sedimentation technique as the concentration-detection method. However, neither of the two studies has dealt with the dissemination of larvae in the milk of naturally infected animals.

Therefore, the current study was set out to detect the experimental and natural infection with *T. vitulorum* larvae in the bovine milk applying two techniques, the classical microscopy-based centrifuge-sedimentation method and the more advanced molecular-based PCR assay.

Additionally, the two methods were compared for their detection limits in both the experimental as well as the natural infection situations.

## Material and methods

### Collection of adult *T. vitulorum* worms

Adult nematodes were collected from the feces of ~ 2-months old, naturally infected buffalo calves with a history of shedding a large number of worms alive. To do so, calves were administered an anthelmintic dosage of piperazine citrate followed by a dosage of magnesium sulphate as a purgative. Six hours later, worms expelled in feces were collected into 0.9% physiological saline in a glass container and were immediately transferred to the laboratory at room temperature [33]. Adult females were differentiated from males under the dissecting microscope (Optika, Italy) based on the differential morphological features [34]. Males were later discarded, while females were maintained as a source of eggs.

### Collection and embryonation of *T. vitulorum* eggs

Adult females were washed twice in 0.9% physiological saline and once in double distilled water to remove any debris. Worms were slit-cut along the uterine regions to extract eggs [35]. Worms were left overnight at room temperature (28°C ± 4°C) in water-filled plastic containers, with the occasional gentle squeezing of worm body by smooth-toothed forceps for the efficient release of eggs. Next morning, representative egg samples were examined under the 100X of the binocular optical microscope (Optika, Italy) to check for the quality of the extracted eggs.

For embryonation, eggs were washed 2–3 times in distilled water by gentle centrifugation (800 rpm for 5 minutes). Eggs were then incubated in distilled water in 50 ml falcon tubes at room temperature (28°C ± 4°C) with the volume of water not exceeding 5:1 that volume of the egg mass. Incubation was continued for an additional 4 weeks with eggs were evaluated periodically for embryonic development under the 100X of the binocular optical microscope.

### Hatching and collection of *T. vitulorum* larvae

To break down egg coatings and freeing larvae, embryonated (larvated) eggs in 2-ml distilled water were subjected to vigorous vortexing for 3–5 minutes in the presence of glass beads. After a pause of 5 minutes at 37 °C, egg mixture was subjected to another round of vortexing for 3–5 minutes until the release of > 90% of larvae was obtained. After leaving to stand for 5 minutes, supernatant (which contains eggshells and debris) was discarded, and the bottom part containing larvae was washed twice through centrifugation (800 rpm for 5 minutes) in distilled water and re-suspension. The recovered larvae were stored refrigerated (+ 4°C) until used for the experimental infection of the milk.

### Experimental contamination and recovery of *T. vitulorum* larvae from milk samples

Formation of larvae aliquots followed by the experimental contamination of milk samples with different doses of larvae was done according to [32], but with minor modifications. Essentially, the recovered *T. vitulorum* larvae were counted on the custom microscope slide (sectioned into 4-mm squares) under the 4X of the optical microscope to obtain a count of #larvae/ml. When necessary, the larvae mixture was diluted 1/10 in distilled water to facilitate larvae counting. Thereafter, concentration ascending aliquots of larvae composed of 1, 5, 10, 20, 50 larvae, and the negative control (no larvae) aliquots in 200 μL distilled water were prepared.

Pasteurized whole bovine UHT milk (Bekhero®, Nestlé) was used in this study, with the volume of milk samples was set at 2-ml for each of the tested larvae aliquot. Aliquots of larvae were then added to milk samples in 15-ml falcon tubes.

To recover larvae from milk samples, samples were subjected to the formalin-ether technique as described by [20,32,36], but with the minor modification of using chloroform instead of ether in the milk degreasing process. After the final centrifugation step, 200 μL aliquots from sediment of different larvae preparations were collected. Aliquots for microscopy examination were processed straightforward. Those for molecular analysis were then stored at –20 °C till used.

### Collection of naturally colostrum/milk samples

Milk/Colostrum samples were collected from 50 animals (26 buffalo and 24 cows) during the postparturient period (Days 1–15) [17,37]. Samples were collected into 15-ml falcon tubes and were adequately labeled. Similar to the experimental samples, 2-ml aliquots were either directly processed for the microscopy examination, or were kept frozen at –20 °C for the molecular examination.

### Microscopy detection

At the end of the formalin-chloroform degreasing of milk samples and the centrifugation step, 200 μL sample of the sediment was transferred to a glass slide (sectioned into 4-mm squares as above), covered with a coverslip, and larvae were counted under the 100X of the optical microscope. Triplicate aliquots from each larvae preparation were examined and larvae were counted, with counting was done by two researchers in a blind-folded mode, giving the total of six independent counts being made from each group. Alongside each of the examined larvae preparation, a 2-ml infection-free milk sample was processed and assessed as a control.

### Molecular detection

From the experimentally-infected and naturally-occurring milk samples, 2-ml milk samples stored at -20 °C were centrifuged, and the 200 μL sediment samples were subjected to the genomic DNA extraction using commercial DNA extraction kits (Intron, South Korea) following the manufacturer instructions. The final extracted genomic DNA was eluted in 50 μL volumes. All samples were uniformly subjected to the PCR amplification reactions. For PCR, a 590-bp fragment of the ITS-1 gene was amplified by the primer pair F2662 (5′–GGCAAAAGTCGTAACAAGGT–3′) and R3214 (5′–CTGCAATTCGCACT ATTTATCG–3′), in 50 μL PCR reactions as done previously [33,38]. Similar to microscopy, a larvae-free control sample was used as negative control.

### Statistical analysis

In the experimental infection, six replicates of different larvae preparations (1, 5, 10, 20, 50 larvae/ml) were counted for larvae presence, with mean, range, and the total numbers were calculated by the Microsoft Excel. Differences between different groups were statistically assessed by the Chi-square test. The two detection methods, microscopy and molecular, were compared by using the unpaired t-test. The two tests were included in the SPSS statistical package (SPSS Statistics for Windows, version x.0, SPSS Inc., Chicago, Ill., USA.), with the level of significance in both tests was set at 5% ($p < 0.05$).

To measure the agreement between microscopy and molecular data, Kappa coefficient was used with range of 0.80–1.00 = perfect agreement, 0.61–0.80 = substantial agreement, 0.41–

0.60 = moderate agreement, 0.21–0.40 = fair agreement, 0.10–0.20 = slight agreement, <
0.10 = poor agreement, as previously described [39].

### Ethical considerations

Collection of samples was performed as a part of the routine animal care, without exerting
unnecessary external stressful factors on animals. Collection of nematode worms was done
during the routine de-worming protocol of the infected calves, while that the milk and colos-
trum samples were collected during the routine milking process.

## Results

### Collection of adult T. *vitulorum* worms

Collected worms were identified as Ascarid nematodes of *T. vitulorum* based on morphologi-
cal characters. Both female and male adult worms were identified based on the morphometric
characters [34].

### Collection and embryonation of *T. vitulorum* eggs

Eggs extracted from adult female *T. vitulorum* worms showed successful embryonic develop-
ment under the laboratory conditions, in which eggs developed from unembryonated stage to
eggs with fully developed embryonic larvae passing through various embryonic developmental
stages that were documented microscopically (Fig 1). Nearly >80% of eggs were detected with
fully developed larvae at the end of the 28-days incubation period (S1 Video). Larvae were
extracted from fully embryonic-developed eggs (Fig 1 #6) by the mechanical breaking (S1 Fig).

### Detection of *T. vitulorum* larvae in experimentally-infected milk samples

With except the control milk sample, microscopic examination detected larvae in all aliquots
preparation. Nevertheless, the mean, range and the total number of larvae recovered recorded
a significant (p<0.05) exponential rate of increase in relative to the ascending larvae concen-
trations (1, 5, 10, 20, 50), as shown in Table 1. Total numbers of the recovered larvae were 2, 3,

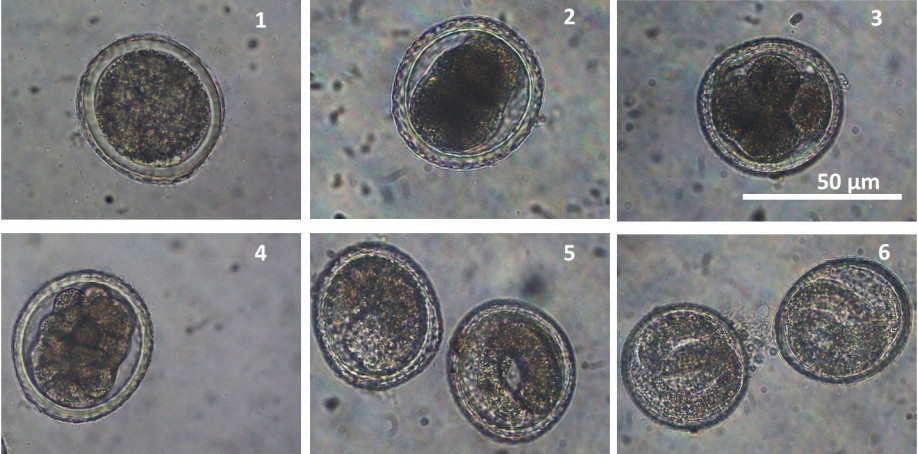

**Fig 1. Developmental progress of *T. vitulorum* eggs from unembryonated stage (1) to eggs with fully developed
embryonic larvae (6), passing through various embryonic developmental stages (2–5).** Stage 1 to 2 = 8 days (d),
2–3 = 6 d, 3–4 = 4 d, 4–5 = 2 d, 5–6 = 2 d.

**Table 1. Detection of larvae of *T. vitulorum* in experimentally contaminated bovine milk samples.** Mean, range, and the total number of larvae detected microscopically from different preparation groups (0, 1, 5, 10, 20, 50 larvae) are indicated. Molecular detection of larvae in different preparations is indicated either negative or positive based on the PCR results.

| Groups (Larvae/2-ml milk aliquot) | Microscopy | | | | Molecular (PCR) |
|---|---|---|---|---|---|
| | Mean* | Range** | Total number of larvae recovered | p-value*** | |
| G0 (0 L), control | 0.00 | (0–0) | 0 | 0.0236 (p<0.05) | Negative |
| G1 (1 L) | 0.57 | (0–1) | 2 | | Positive |
| G2 (5 L) | 0.86 | (0–1) | 3 | | Positive |
| G3 (10 L) | 1.71 | (0–3) | 6 | | Positive |
| G4 (20 L) | 4.00 | (1–6) | 14 | | Positive |
| G5 (50 L) | 5.43 | (1–6) | 19 | | Positive |

* Average number of larvae that were recovered from the sample count in 6 replicates

** Range of larvae recovered across the 6 replicates of each sample

*** Level of significance was calculated out from the sample replicates by the Chi-square test, where p<0.05 indicated significant differences

6, 14, and 19, in groups 1–5, respectively. On the other hand, PCR detected the genomic DNA originating from larvae in all preparations, irrespective of larvae concentration (Table 1). Again, a negative PCR result was documented in the control (no larvae) milk sample.

## Detection of *T. vitulorum* larvae in naturally-occurring milk samples

As shown in Table 2, a total of 50 bovine colostrum/milk samples were examined for the presence of larvae by microscopy examination, and from which 13 (26%) were tested positive. When dissected, 11 out of 26 buffalo samples (42.30%) and 2 out of 24 cow samples (8.33%) were positive. When the same samples (50) were subjected to DNA extraction and the PCR detection, a more sensitive detection was obtained, with 16 (61.54%) and 3 (12.50%) of buffalo and cow samples were tested positive, respectively (Fig 2; Table 2). Not every microscopy-positive sample was also PCR-positive. However, substantial agreement (0.77 Agreement Coefficient) between molecular and microscopy detection was detected from all tested samples (Table 2). This is dissected into a perfect agreement with buffalo samples (0.82) and a moderate agreement (0.5) with cow samples.

## Discussion

Transmammary transmission of *T. vitulorum* from infected dams to the newborn calves constitutes a challenge to the livestock keepers, particularly when toxocariasis is widespread, and when anthelmintic treatment are not routinely applied [6,7,10]. Several techniques were applied in detecting larvae of *Toxocara* species in the milk of infected mothers. Microscopy-based detection applying the centrifuge-sedimentation technique and using the formalin-ether

**Table 2. Microscopy versus molecular detection of naturally occurring infection of *T. vitulorum* in the milk of bovine hosts.**

| Host | Number | Microscopy Number/Percent | Molecular Number/Percent | Agreement Coefficient (Molecular/Microscopy) | p-value |
|---|---|---|---|---|---|
| Buffalo | 26 | 11 (42.30) | 16 (61.54) | 9/11 (0.82; Perfect) | 0.0568 |
| Cow | 24 | 2 (8.33) | 3 (12.50) | 1/2 (0.5; Moderate) | |
| Total | 50 | 13 (26.00) | 20 (40.00) | 10/13 (0.77; substantial) | |

**Agreement Coefficient**: 0.80–1.00 = perfect agreement, 0.61–0.80 = substantial agreement, 0.41–0.60 = moderate agreement, 0.21–0.40 = fair agreement, 0.10–0.20 = slight agreement, < 0.10 = poor agreement

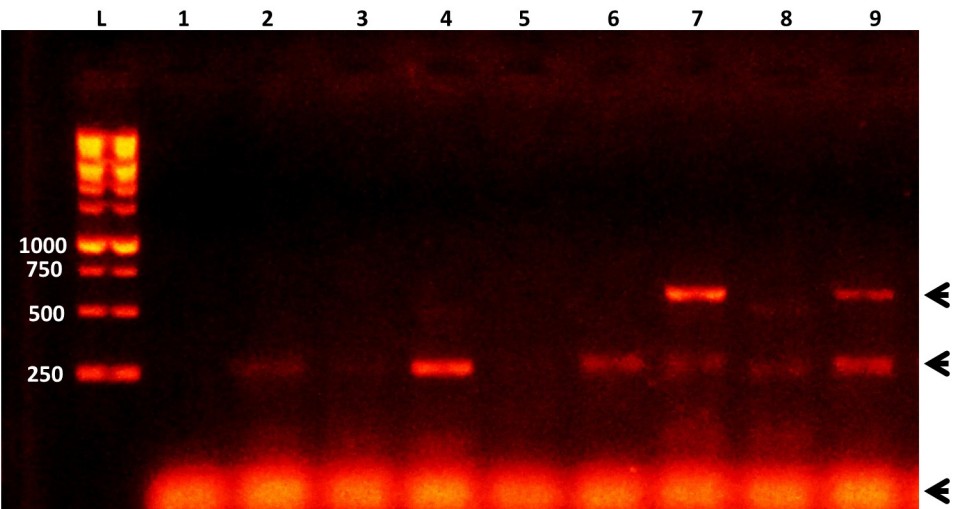

**Fig 2. A representative gel of PCR-based detection of *T. vitulorum* in colostrum milk samples of bovine animal hosts.** L = DNA ladder, 1 = Negative control, 2 = Milk sample control, 3–9 = Colostrum milk samples. DNA ladder sizes (in bp) are indicated to the left. Arrows heads indicate either the positive PCR reaction (upper), the non-specific band (middle), or the primer-dimer band (bottom).

solution is considered as an easy, cheap and non-time wasting technique with the minimum detection limit of one larva of *T. canis* in milk [32]. Currently, DNA techniques have been widely used in the identification and genetic differentiation of *T. vitulorum* and other *Toxocara* species [40,41]. In our present study, larvae of *T. vitulorum* were successfully recovered from milk samples both after experimental contamination and from colostrum samples of naturally infected animals. In both cases, a highly sensitive detection limit was obtained that went down to a single larva per sample. Nevertheless and when compared side by side, PCR tends to be more sensitive than microscopy in detecting naturally-infected milk samples.

Thus two techniques could be used in conjunction for the successful recovery of larvae from the suspected milk samples. When the microscopic examination returned negative results, PCR could be successfully applied in the molecular detection of parasites. PCR has been applied unequivocally for the detection of animal and human toxocariasis. An example was the molecular detection of *Toxocara* species eggs in the soil-tainted and faecal samples of dogs and cats [40,42,43]. Concerning human toxocariasis, PCR-based molecular detection was applied in the examination of experimental samples of bronchoalveolar lavage and the cerebrospinal fluid (CSF) samples from patients with neurological symptoms [44,45], and from a patient with visceral larva migrans lesions in his eyes [46].

When compared side-by-side, the two detection methods were proved sensitive enough to detect a single larva in the 2-ml milk sample. The molecular technique would fit more the research purposes as it deemed non-practical for the in-field diagnostic approach. On the other hand, detection through concentration-sedimentation and the microscopic examination would constitute a golden standard for surveying the milking animals. This, in turn, would allow the exclusion and subsequent treatment of infected dams, with providing an alternative milk source to suckling claves. [47] found that the treatment of *T. vitulorum* infection based essentially on the duration of the infectivity of buffalo cows for their calves. If the infection is prevalent in the herd, artificial milking feeders with pasteurized milk would be the optimum alternative.

Our data on the experimental contamination and detection of *T. vitulorum* larvae in milk samples are in close agreement with that previously recorded with other *Toxocara* species, specifically to name, *T. canis* [32]. The only difference is that this later study compared the effects of different degreasing solutions on milk samples and whether milk was whole or skimmed, factors which they found had no significant effects on the detection limits [32].

Our data on natural infection agree with that obtained by [17,48]. In these studies, larvae of *T. vitulorum* were identified in the colostrum/ milk samples of infected buffalo cows. Larvae were particularly detected with significant numbers during the first two weeks (3–12 days) post parturient. Thus together these data concluded that claves were prone to *T. vitulorum* infection since the first few days of suckling period, and as also confirmed previously [49]. All these confirm that the transmammary route is the principal mode of infection in the case of bovine toxocariasis. Nevertheless, we cannot exclude the prenatal (transplacental) route as a second most important infection source, particularly when larvae of *T. vitulorum* were detected in large numbers in the fetal liver and lungs of the pregnant buffaloes [1].

In addition to microscopy and molecular methods, serology-based detection of anti-*T. vitulorum* antibodies have been used previously to monitor the infection status of buffalo calves and cows [50,51]. To identify and exclude the infected buffalo cows from feeding their newborn calves, the in situ detection of larvae of *T. vitulorum* in colostrum/milk samples as applied in the current study constitutes a more accurate and valid way than the antibody-based serological detection, in which the recent active and the old inactive infections cannot be differentiated [52].

The method applied in this study for the *in vitro* hatching and the release of *T. vitulorum* larvae has been much less complicated and has involved much less mechanical disruption than previously used methods [53,54]. The integrity and viability of freshly obtained larvae were kept in most cases. With the experimental processing, we observed that in most samples, either experimentally or naturally infected, larvae were motionless with some effects on integrity. This is most probably due to the action of formalin and chloroform, chemicals which were used in the larvae recovery process [32].

## Conclusions

We conclude that the applied techniques in the current study could constitute routine detection methods for *Toxocara* larvae in milk from suspected infected animals, particularly in geographic areas with histories of endemic toxocariasis. In the same endemic areas, a standard protocol should be routinely practiced that involve deworming of calves during the early suckling period (2–4) weeks of age.

From the public health perspective, treatment of colostrum/milk from recently parturated animals before human consumption should be considered as a matter of public health importance to minimize the risk of human infection with *T. vitulorum*.

## Supporting information

**S1 Video. Shows fully-embryonated *T. vitulorum* ova with fully formed, viable larvae.**
(MOV)

**S1 Fig. Larvae were extracted from fully embryonic-developed eggs by the mechanical breaking.**
(TIF)

**S2 Fig. The original underlying image for gel data reported in our submission (S1 Raw images).**
(TIF)

## Acknowledgments

We thank several anonymous local buffalo and cattle herds keepers, who willingly allowed the collection of colostrum/milk samples from their newly parturated animals.

## Author Contributions

**Conceptualization:** Amira Dewair, Mohamed Bessat.

**Data curation:** Amira Dewair, Mohamed Bessat.

**Formal analysis:** Amira Dewair, Mohamed Bessat.

**Funding acquisition:** Amira Dewair, Mohamed Bessat.

**Investigation:** Amira Dewair, Mohamed Bessat.

**Methodology:** Amira Dewair, Mohamed Bessat.

**Project administration:** Amira Dewair, Mohamed Bessat.

**Resources:** Amira Dewair, Mohamed Bessat.

**Software:** Amira Dewair, Mohamed Bessat.

**Supervision:** Amira Dewair, Mohamed Bessat.

**Validation:** Amira Dewair, Mohamed Bessat.

**Visualization:** Amira Dewair, Mohamed Bessat.

**Writing – original draft:** Amira Dewair, Mohamed Bessat.

**Writing – review & editing:** Amira Dewair, Mohamed Bessat.

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
