## [Decision Letter · Decision Letter 0]

25 Mar 2020

PONE-D-19-34351

Molecular and microscopic detection of natural and experimental infections of Toxocara vitulorum in bovine milk

PLOS ONE

Dear Dr. Mohamed Bessat

Thank you for submitting your manuscript to PLOS ONE. After careful consideration, we feel that it has merit but does not fully meet PLOS ONE’s publication criteria as it currently stands. Therefore, we invite you to submit a revised version of the manuscript that addresses the points raised during the review process.

We would appreciate receiving your revised manuscript by May 09 2020 11:59PM. To enhance the reproducibility of your results, we recommend that if applicable you deposit your laboratory protocols in protocols.io, where a protocol can be assigned its own identifier (DOI) such that it can be cited independently in the future. For instructions see: http://journals.plos.org/plosone/s/submission-guidelines#loc-laboratory-protocols

We look forward to receiving your revised manuscript.

Kind regards,

Jacopo Guccione

Academic Editor

PLOS ONE

Journal Requirements:

"The study was in-part supported financially by the Vmerge FP7 grant (613996) from the EU. The funder had no role in study design, data collection and analysis, decision to publish, or preparation of the manuscript. "

Reviewers' comments:

Reviewer's Responses to Questions

**Comments to the Author**

1. Is the manuscript technically sound, and do the data support the conclusions?

Reviewer #1: Yes

Reviewer #2: Yes

2. Has the statistical analysis been performed appropriately and rigorously? 

Reviewer #1: Yes

Reviewer #2: Yes

3. Have the authors made all data underlying the findings in their manuscript fully available?

Reviewer #1: Yes

Reviewer #2: Yes

4. Is the manuscript presented in an intelligible fashion and written in standard English?

Reviewer #1: No

Reviewer #2: Yes

5. Review Comments to the Author

Reviewer #1: The manuscript PONE-D-19-34351 reports the results of a study aimed at detecting the experimental and natural infection with Toxocara vitulorum larvae in the bovine milk applying two techniques, i.e. the classical microscopy-based centrifuge-sedimentation method and a molecular-based PCR assay.

The experimental set-up permitted to recover larvae of T. vitulorum from milk samples both after experimental contamination and from colostrum samples of naturally infected animals.

I recommend the publication of the paper in Parasites and Vectors after major revisions. The questions posed by the authors are well defined, the methods appropriate and the results, discussion and conclusions well balanced and adequately supported by the data.

However, there some issue which should be addressed by the Authors.

• Introduction is confusing and with too many paragraphs on Toxocara canis and T. cati (e.g. lines 62 to 72)

• In the introduction, I would suggest to provide prevalence of T. vitulorum in different geographical areas

• Figure 1 is not necessary to the MS. I would suggest to remove it.

• Line 190. I think that “Mancianti et al., 2013” is not an appropriate reference for Kappa coefficient ranking. I suggest the authors to use a statistical reference.

• In the discussion the authors mention microscopy and molecular methods for detection of larvae in milk samples, as well as antibody-based serological methods. What about antigen-based milk ELISA methods? And what about copromicroscopic methods?

• Furthermore, do the author have any clue regarding the costs of each diagnostic method?

Reviewer #2: Ref: PONE-D-19-34351

Title: Molecular and microscopic detection of natural and experimental infections of Toxocara vitulorum in bovine milk

The objective of this study was to detect the experimental and natural infection with Toxocara vitulorum larvae in the bovine milk applying two techniques, the classical microscopy-based centrifuge-sedimentation method and the more advanced molecular-based PCR assay.

The manuscript is well written, and the results are very interesting.

The paper is accepted after minor revision.

However, see comments below:

Abstract

Line 11: replace “T. vitulorum” with “Toxocara vitulorum”

Introduction

Line 41: Please, put T. vitulorum between two brackets and remove the commas.

Material and Methods

Lines 96-106: Insert the bibliographic reference of the method used.

Lines 106-119: Insert the bibliographic reference of the method used.

Lines 119-129: Insert the bibliographic reference of the method used.

Line 151: Explain if the 50 animals have undergone copromicroscopic examination to know their positivity to T. vitulorum.

Lines 232-234: add for each stage the time (days) of development of the larvae in the egg.

Line 234: In the legend of Fig. 2, remove "Scale bar is 50 μm" and add "50 μm" on the white bar in the photos.

Results

Line 284: Add a photo of larvae identified with the optical microscope.

Discussion

Lines 297-298: The bibliographic references cited are very outdated, use or add more updated ones.

Line 358: cursive writing in vitro.

General comments:

The overall paper is well written, but I think you should add more bibliographic references regarding the methods used in laboratory tests.

Moreover, bibliographic references of the introduction and discussion should be more up to date.

6. PLOS authors have the option to publish the peer review history of their article (what does this mean?). If published, this will include your full peer review and any attached files.

Reviewer #1: No

Reviewer #2: No

---

## [Author Response · Author response to Decision Letter 0]

6 Apr 2020

Reviewer #1:

The manuscript PONE-D-19-34351 reports the results of a study aimed at detecting the experimental and natural infection with Toxocara vitulorum larvae in the bovine milk applying two techniques, i.e. the classical microscopy-based centrifuge-sedimentation method and a molecular-based PCR assay.

The experimental set-up permitted to recover larvae of T. vitulorum from milk samples both after experimental contamination and from colostrum samples of naturally infected animals.

I recommend the publication of the paper after major revisions. The questions posed by the authors are well defined, the methods appropriate and the results, discussion and conclusions well balanced and adequately supported by the data. However, there some issue which should be addressed by the Authors.

• Introduction is confusing and with too many paragraphs on Toxocara canis and T. cati (e.g. lines 62 to 72)

Response: This is due to the matter of fact that the dog and cat Toxocara species are the most studied, and to pin down the fact that T. vitulorum did received the least attention despite its economic and medical importance. Thus it highlights the importance of our study, and why it was initiated.

• In the introduction, I would suggest to provide prevalence of T. vitulorum in different geographical areas

Response: A paragraph describes the prevalence of T. vitulorum in different geographical areas has since been added to the “Introduction” section (Lines: 48-55). 

• Figure 1 is not necessary to the MS. I would suggest removing it.

Response: Figure 1 has since been removed, as per our agreement with the reviewer’s recommendation.

• Line 190. I think that “Mancianti et al., 2013” is not an appropriate reference for Kappa coefficient ranking. I suggest the authors to use a statistical reference.

Response: A more specialized statistical reference has been applied to cite the statistical method of “Kappa coefficient”. I would thank the reviewer for pinpointing the correction.

• In the discussion the authors mention microscopy and molecular methods for detection of larvae in milk samples, as well as antibody-based serological methods. What about antigen-based milk ELISA methods? And what about copromicroscopic methods?

Response: Unfortunately and after deep and rigorous mining of all publications on T. vitulorum, only antibody-based serological methods had been used to monitor excretion of parasite in the colstrum/milk samples, with no data on the antigen-based milk ELISA. Also since our study was only focused on the milk excretion with no interest of faecal excretion data, the copromicroscopic methods become of less significance to be mentioned in the “Discussion” part. But again I would appreciate the reviewer for raising the point here. 

• Furthermore, do the author have any clue regarding the costs of each diagnostic method?

Response: In short, the PCR-based molecular method would be more expensive than the microscopic examination after the concentration-sedimentation technique (would approximate 3000 Egyptian pound/180 USD per 50 samples versus ~ 800 EGP/50 USD, respectively). Albeit, the molecular method still to give more sensitive detection levels when compared with the microscopy method when applied in the experimental part of the study. Thus, and again as mentioned in the “Discussion” part, the molecular technique would fit more the research purposes, while the microscopic examination would constitute a golden standard for surveying the milking animals. 

I shall thank the reviewer for recommending these constructive corrections, which with no doubt have added much to the manuscript, leading to much improvement in its quality and readability.

Reviewer #2

Title: Molecular and microscopic detection of natural and experimental infections of Toxocara vitulorum in bovine milk.

The objective of this study was to detect the experimental and natural infection with Toxocara vitulorum larvae in the bovine milk applying two techniques, the classical microscopy-based centrifuge-sedimentation method and the more advanced molecular-based PCR assay.

The manuscript is well written, and the results are very interesting.

The paper is accepted after minor revision.

However, see comments below:

o Abstract: 

• Line 15: replace “T. vitulorum” with “Toxocara vitulorum”

Response: It has been corrected

o Introduction: 

• Line 43: Please, put T. vitulorum between two brackets and remove the commas.

Response: It has been corrected

o Material and Methods

• Lines 96-106: Insert the bibliographic reference of the method used.

Response: A suitable bibliographic reference of the method used has been inserted.

• Lines 106-119: Insert the bibliographic reference of the method used.

Response: The method used was based solely on our trials and observations during the pilot and preliminary experiments in the Lab. It has been written in enough details so giving the fellow researchers the opportunity of copying it into their Labs. 

• Lines 119-129: Insert the bibliographic reference of the method used.

Response: The method used was based solely on our trials and observations during the pilot and preliminary experiments in the Lab. It has been written in enough details so giving the fellow researchers the opportunity of copying it into their Labs. 

• Line 151: Explain if the 50 animals have undergone copromicroscopic examination to know their positivity to T. vitulorum.

Response: No, Milk/Colostrum samples were collected blindly from 50 animals, without performing any copromicroscopic examination. This is because the main specific aim was to compare the same two methods blindly on the same milk samples, as it is not necessarily that the same copro-positive animal to excrete the parasite in its milk. 

• Lines 232-234: add for each stage the time (days) of development of the larvae in the egg.

Response: Information about developmental periods (in days) has been added.

• Line 234: In the legend of Fig. 2, remove "Scale bar is 50 μm" and add "50 μm" on the white bar in the photos.

Response: It has been corrected on the photos.

o Results

• Line 284: Add a photo of larvae identified with the optical microscope.

Response: Supporting files (Supporting video and figure) have been added to show the viable larvae inside the fully-embryonated eggs, as well as larvae being extracted from mechanically-broken eggs. 

o Discussion

• Lines 297-298: The bibliographic references cited are very outdated, use or add more updated ones.

Response: More updated references have been added accordingly, many thanks for the reviewer for his recommendation.

• Line 358: cursive writing in vitro.

Response: It has since been corrected.

o General comments:

The overall paper is well written, but I think you should add more bibliographic references regarding the methods used in laboratory tests.

Moreover, bibliographic references of the introduction and discussion should be more up to date.

Response: As it can be easily figured out that the parasite of T. vitulorum is considered as a neglected when compared to the heavily-researched counterparts of dog and cat ascarids, and thus just barely over 20 papers had been published throughout the last decade. Nevertheless, more updated references have been added to the introduction and discussion sections, to reflect upon the reviewer’s recommendations. 

We finally would greatly appreciate the reviewer for his comments which led to much improvement in the quality of the manuscript.

 In all, we consider the paper much improved after the suggested revisions, and would like to thank you and the reviewers for your kind help. We do hope that the paper is now deemed acceptable for publication in PLoS One.

---

## [Decision Letter · Decision Letter 1]

6 May 2020

Molecular and microscopic detection of natural and experimental infections of Toxocara vitulorum in bovine milk

PONE-D-19-34351R1

Dear Dr. M Basset,

We are pleased to inform you that your manuscript has been judged scientifically suitable for publication and will be formally accepted for publication once it complies with all outstanding technical requirements.

With kind regards,

Jacopo Guccione

Academic Editor

PLOS ONE

Additional Editor Comments (optional):

Reviewers' comments:

Reviewer's Responses to Questions

**Comments to the Author**

1. If the authors have adequately addressed your comments raised in a previous round of review and you feel that this manuscript is now acceptable for publication, you may indicate that here to bypass the “Comments to the Author” section, enter your conflict of interest statement in the “Confidential to Editor” section, and submit your "Accept" recommendation.

Reviewer #1: All comments have been addressed

Reviewer #2: All comments have been addressed

2. Is the manuscript technically sound, and do the data support the conclusions?

Reviewer #1: Yes

Reviewer #2: Yes

3. Has the statistical analysis been performed appropriately and rigorously? 

Reviewer #1: Yes

Reviewer #2: Yes

4. Have the authors made all data underlying the findings in their manuscript fully available?

Reviewer #1: No

Reviewer #2: Yes

5. Is the manuscript presented in an intelligible fashion and written in standard English?

Reviewer #1: Yes

Reviewer #2: Yes

6. Review Comments to the Author

Reviewer #1: The authors have addressed the issues required by the reviewers. The manuscript can be accepted for publication.

Reviewer #2: Dear Authors,

you have performed all required comments and congratulations on your paper.

7. PLOS authors have the option to publish the peer review history of their article (what does this mean?). If published, this will include your full peer review and any attached files.

Reviewer #1: Yes: Laura Rinaldi

Reviewer #2: No

---

## [Editor Report · Acceptance letter]

11 May 2020

PONE-D-19-34351R1 

Molecular and microscopic detection of natural and experimental infections of *Toxocara vitulorum* in bovine milk 

Dear Dr. Bessat:

I am pleased to inform you that your manuscript has been deemed suitable for publication in PLOS ONE. Congratulations! Your manuscript is now with our production department. 

With kind regards,

on behalf of

Dr. Jacopo Guccione 

Academic Editor

PLOS ONE